# Synergistic Correlation in the Colloidal Properties of TiO2 Nanoparticles and Its Impact on the Photocatalytic Activity

**María Concepción Ceballos-Chuc** [1,2,*] , **Carlos Manuel Ramos-Castillo** [2,3], **Manuel Rodríguez-Pérez** [1,*] , **Miguel Ángel Ruiz-Gómez** [2] , **Geonel Rodríguez-Gattorno** [2] and **Julio Villanueva-Cab** [4]

1 Facultad de Ingeniería, Universidad Autónoma de Campeche, San Francisco de Campeche, Campeche 24085, Mexico
2 Department of Applied Physics, CINVESTAV-IPN, Antigua Carretera a Progreso Km 6, Merida 97310, Yucatan, Mexico
3 División de Investigación y Posgrado, Universidad Autónoma de Querétaro, Querétaro 76010, Mexico
4 Instituto de Física, Ecocampus-Valsequillo, Benemérita Universidad Autónoma de Puebla, Puebla 72570, Mexico
* Correspondence: mcceball@uacam.mx (M.C.C.-C.); mjrodrig@uacam.mx (M.R.-P.)

**Abstract:** In this work, the relationship between the photodegradation rate of methylene blue (MB) and the effective surface charge of titania nanoparticles (TiO2 NPs) in an aqueous solution is addressed. Colloidal dispersions were prepared from TiO2 NPs (4–10 nm) for the heterogenous photocatalysis test. The dispersion properties such as pH, hydrodynamic diameter, zeta potential, and isoelectric point were studied. Acidic TiO2 dispersions (pH = 3.6–4.0) with a positive zeta potential and smaller hydrodynamic diameter exhibit larger colloidal stability and pseudo-first-order kinetics for the degradation of MB. The largest rate constant ($5 \times 10^{-2}$ min$^{-1}$) corresponded to a conversion of 98% within 75 min under UV light. This enhanced rate is a synergic effect between the surface area, charge, and optimal hydrodynamic diameter of TiO2 NPs. A linear correlation between the calculated values for the absorption cross-section and normalized rate constant was found for the systems under study. It was observed that an eventual increase in the pH (4–5.5) reduces the effective surface charge and dispersion stability, causing a decrease in the rate constants of one order of magnitude ($10^{-3}$ min$^{-1}$) for TiO2 agglomerates with a larger hydrodynamic diameter (300–850 nm).

**Keywords:** colloidal TiO2 dispersion; surface charge; photocatalysis

## 1. Introduction

Due to their excellent physical and chemical properties, titanium dioxide nanoparticles (TiO2 NPs) often are used for environmental cleanup applications to degrade pollutants in water [1–8]. The structural properties of TiO2 such as the crystal structure, morphology, particle size, bandgap, and surface area play a key role in determining its photocatalytic properties [1,9–20]. However, due to several photoinduced applications requiring the dispersion of TiO2 NPs in colloidal media, the understanding of how factors such as the surface charge and pH of the dispersion can regulate the photodegradation rate is mandatory [9,10,21–25].

The pH of the solution is a key factor affecting the photodegradation rate of pollutants and dyes. The generation of different reactive oxygen species (ROS) able to degrade pollutants and dyes is related to the pH of the dispersion and the surface charge of the nanomaterials [26–28]. Different strategies have been implemented to achieve larger photodegradation rates, such as surface doping, heterojunctions, co-catalysts, increasing the surface area, and highly reactive facets [29,30]. The pH of the solution influences the electrical double layer of the solid–electrolyte interface and consequently affects the generation and separation of the electron-hole pairs. For TiO2 NPs, either a positive or a negative charge can be developed on its surface, which has an influence on the adsorption

of molecules [31]. In this respect, the photodegradation of cationic methylene blue (MB$^+$) in TiO$_2$ dispersions has received special attention. The effect of pH on photodegradation has been explained on the basis of an electrostatic model. This model establishes that in acidic conditions (pH < 5), the surface of TiO$_2$ NPs is positively charged and electrostatic repulsion leads to a decrease in the molecular adsorption of cationic dyes on the TiO$_2$ surface, resulting in a lower degradation efficiency [32]. In contrast, basic conditions are expected to increase the molecular adsorption on the surface of negatively charged nanoparticles, enhancing the rate of the photocatalytic discoloration of methylene blue (MB) [33]. The basic idea behind the electrostatic model is that only adsorbed molecules are available to be attacked by photogenerated holes and hydroxyl radicals on the catalyst surface and, therefore, only these contribute to the photodegradation. This has caused most of the research on the photocatalysis of MB in TiO$_2$ to be conducted under basic conditions. Kim et al. [34] have reported that anatase nanoparticles can generate mobile hydroxyl radicals that can diffuse to the solution and react with both adsorbed and non-adsorbed dye molecules and is a more versatile oxidant. In this respect, Harris et al. [35] have reported the synthesis and photocatalytic activity of hierarchical anatase nanoflowers. A degradation rate 14 times faster than that of P25 TiO$_2$ at pH 4 was measured, representing one of the highest activities yet reported for MB photo-oxidation with TiO$_2$.

Other factors related to the pH and photocatalytic activity are the absorption and scattering processes of light by dispersed NPs. Nanoparticles in dispersion often form aggregates or agglomerates. The change in the pH can alter the colloidal stability of the dispersed NPs, affecting the light absorption and scattering, with dramatic consequences on the photodegradation rate [36–38]. The formation of agglomerates is related to the decrease in the net surface charge of dispersed NPs. In photocatalysis, light absorption and scattering are influenced by the size of agglomerates, because UV photons have a wavelength (~300 nm) much larger than the typical spaces between the primary particles (10–20 nm). Consequently, it is expected that changes in the degree of particle agglomeration, such as those brought about by milling, agitation, or even pH, should change the measured photocatalytic activity. However, there are only a few reports addressing the relationship between the pH, light absorption, and photocatalytic performance of dispersed TiO$_2$ NPs [39–41]. To address this issue, the relationship between the photocatalytic activity and the optical response of TiO$_2$ NPs in dispersion was investigated in the framework of the Mie scattering theory in combination with the measurement of dynamic light scattering (DLS).

In this study, the characterization of TiO$_2$ NPs in deionized water (DI-water) and DI-water/MB dispersions was performed using DLS. The influence of the initial pH, hydrodynamic diameter (D$_H$), and particle surface chemistry ($\zeta$-potential) on the photocatalytic activities of as-prepared powders was studied toward the degradation of aqueous MB as a model pollutant. The calculated absorption and scattering cross-sections as a function of D$_H$ predict that for the systems under study, the light scattering is not negligible in comparison with the light absorption. The results show a direct correlation between the photocatalytic degradation rate constant and the total adsorption cross-section of TiO$_2$ NP dispersions, which indicates that the optical adsorption and D$_H$ of the agglomerates are key factors that determine the photocatalytic properties of semiconductor oxides in dispersion.

## 2. Results and Discussion

### 2.1. Crystal Structure and Morphology of TiO$_2$ NPs

The X-ray diffraction (XRD) patterns of the TiO$_2$ NP photocatalyst prepared by microwave heating method are shown in Figure 1a. The observed diffraction peaks correspond to the expected reflections for the anatase crystal phase (JCPDS # 21-1272), and a small fraction (~10%) of the brookite phase was detected. Raman spectroscopy was also used for phase identification analyses. The measured Raman spectra are presented in Figure A1, showing that the TiO$_2$ NPs crystallized principally in the anatase phase exhibit the six Raman active modes with symmetry E$_g$ (144 cm$^{-1}$), E$_g$ (196 cm$^{-1}$), B$_{1g}$ (397 cm$^{-1}$), doublet A$_{1g}$ + B$_{1g}$ (516 cm$^{-1}$), and E$_g$ (640 cm$^{-1}$). Moreover, very low intensity peaks at A$_{1g}$

(246 cm$^{-1}$) and B$_{1g}$ (322 cm$^{-1}$) [42,43] reveal the presence of a small quantity of the brookite phase, which is in good agreement with the XRD results.

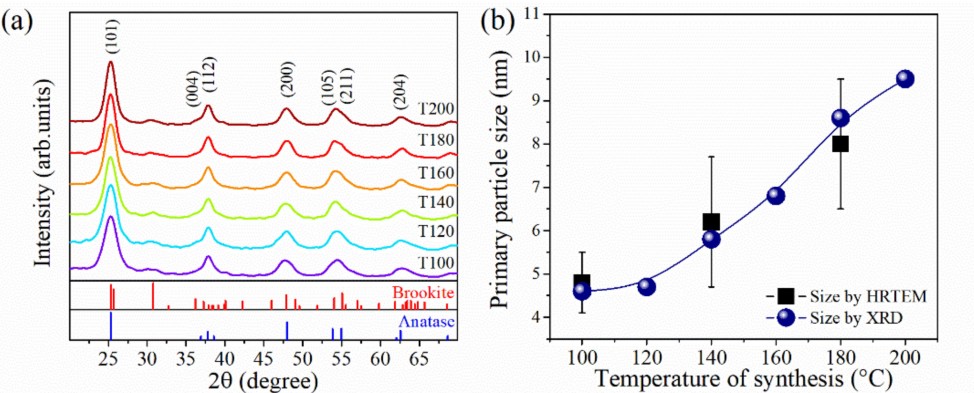

**Figure 1.** (**a**) Evolution of the XRD patterns of samples obtained at 100–200 °C. (**b**) Comparison of the mean primary particle size determined by XRD and HRTEM analysis for all synthesized TiO$_2$ NPs. Representative HRTEM images and size distribution analysis are shown in Figure A2.

The mean crystallite size (primary particle size) was calculated with Bruker's program Topas-4 by fitting all the peaks. It is observed that the size of the TiO$_2$ NPs increases with the synthesis temperature (Figure 1b). This dependence is expected because the particle growth rate is related to the temperature [21,44].

A series of selected samples were studied by high-resolution transmission electron microscopy (HR-TEM). Some selected micrographs of samples obtained at 100, 140, and 180 °C are shown in Figure A2. The inset plots correspond to the particle size distribution; note that the values of the mean particle size are consistent with those estimated by the XRD analysis (Figure 1b). The observation of samples at low magnification suggests the existence of secondary particles as a result of agglomeration; a further study was accomplished by DLS, and the results are shown hereafter. Figure 2 shows representative SEM (scanning electron microscope) images of all synthesized samples. It can be clearly seen that the nanoparticle size increases with the increase in temperature, which is absolutely in agreement with the XRD results. The morphology of semi-spherical nanoparticles forming agglomerates was observed.

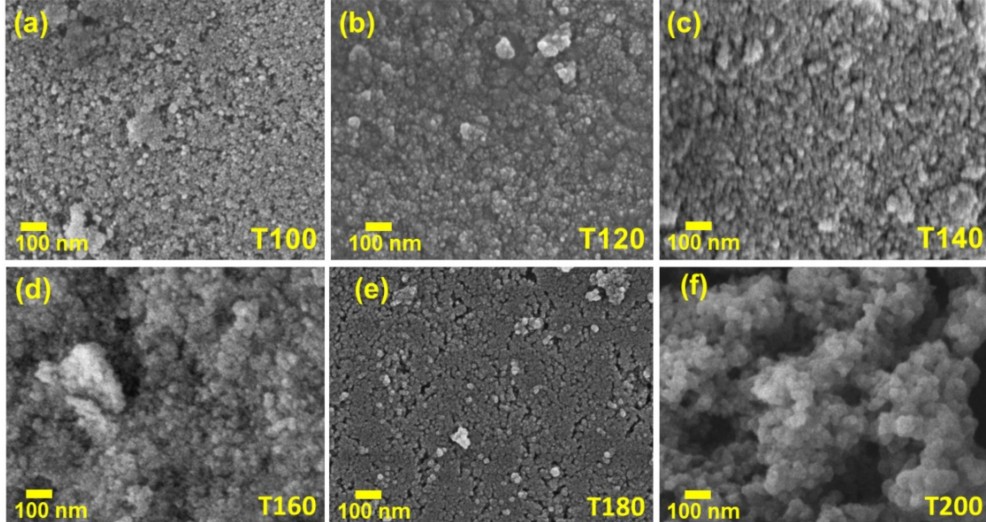

**Figure 2.** SEM images of TiO$_2$ NPs at different microwave hydrothermal temperatures, (**a**) 100 °C, (**b**) 120 °C, (**c**) 140 °C, (**d**) 160 °C, (**e**) 180 °C, and (**f**) 200 °C.

*2.2. Elemental Analysis and Textural Surface Characterization of TiO$_2$ NPs*

The chemical species on the surface as well as their respective oxidation states was determined by X-ray photoelectron spectroscopy (XPS) (see Figure 3). Deconvolution (dash black lines presented of the spectra) shows that the C 1s spectra (Figure 3a) show three peaks at 284.6 eV (C-C bond), 286.2 eV (C-OH), and 288.6 eV (C=O, carboxylic species) [45]. In Figure 3b, the O 1s peak shows three kinds of oxygen; the peak at 530.14 eV is attributed to Ti (IV)-O bonds [46], the peak at 530.6 eV can be attributed to chemisorbed hydroxyl groups (Ti-OH) [47], and the peak at 531.5 eV is attributed to C=O [48]. The chemical bonding states of the Ti 2p$_{3/2}$ peaks were also investigated; Figure 3c shows the Ti 2p core level spectra. The deconvolution of the Ti 2p peaks yields two major doublets (2p$_{3/2}$ and 2p$_{1/2}$) appearing at 458.8 eV and 464.6 eV, respectively, which can be assigned to Ti$^{4+}$ 2p of TiO$_2$, that are typical for anatase [45,46]. The Ti-OH suggests the formation of hydroxyl groups on the surface of TiO$_2$. The surface -OH groups contribute to an increase in the hydrophilicity and enhanced binding of organic molecules [49]. During the photocatalytic process, photoinduced holes react with these groups to yield surface hydroxyl radicals that have a high oxidation capability [21,50].

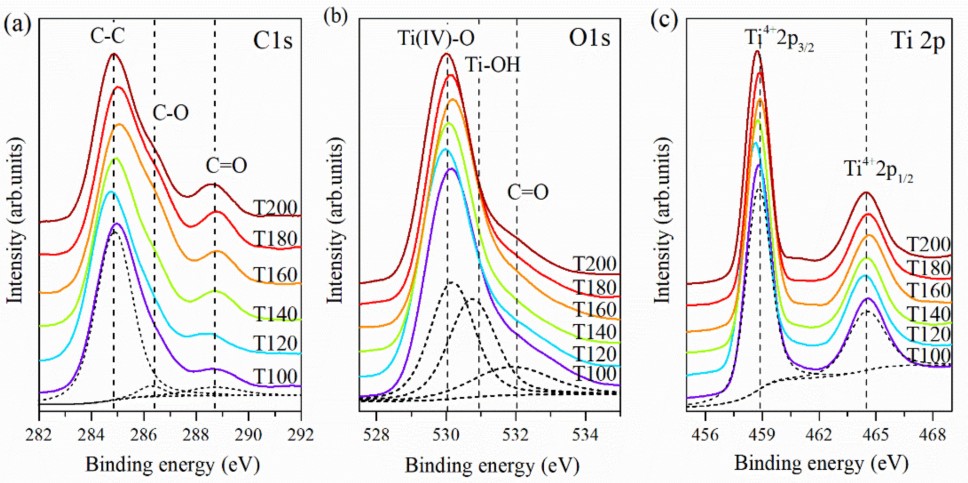

**Figure 3.** XPS spectra for all TiO$_2$ NP photocatalyst samples. Deconvoluted peaks are presented as dash black lines: (**a**) C 1s, (**b**) O 1s, and (**c**) Ti 2p.

On the other hand, the textural properties are summarized in Table 1, while the adsorption–desorption curves are shown in Figure 4a. The isotherm of the T100 and T120 samples corresponds to isotherm type I, indicating their microporous structure with relatively small surfaces. The isotherms of the T140, T160, T180, and T200 samples are type IV, suggesting the presence of a mesoscale porosity [44]. These samples have the well-developed hysteresis loop for P/Po $\geq$ 0.45, signifying the existence of compact agglomerates of cylindrical-like pores [51]. Due to the micro and mesoporous structure of the TiO$_2$ NP samples, they can be considered excellent photocatalysts because it is a desirable condition to improve photocatalysis processes [18].

**Table 1.** Specific surface area S$_{BET}$, pore volume, and pore size of TiO$_2$ NP samples.

| Photocatalyst | S$_{BET}$ (m$^2$ g$^{-1}$) | Pore Volume (cm$^3$ g$^{-1}$) | Mean Pore Diameter (nm) |
|---|---|---|---|
| T100 | 229 | 0.118 | 2.07 |
| T120 | 230 | 0.119 | 2.06 |
| T140 | 260 | 0.165 | 2.53 |
| T160 | 247 | 0.199 | 3.22 |
| T180 | 233 | 0.232 | 3.97 |
| T200 | 216 | 0.239 | 4.41 |

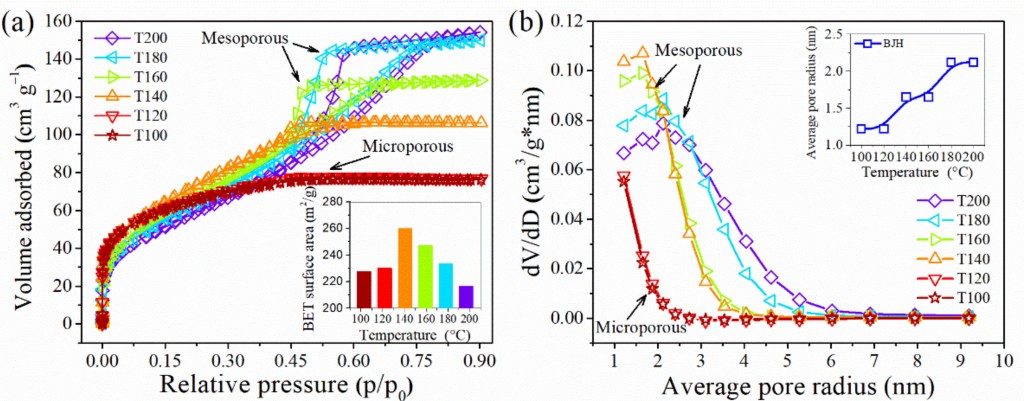

**Figure 4.** (**a**) Brunauer-Emmett–Teller (BET) N$_2$ adsorption–desorption curves for TiO$_2$ NP photocatalyst samples (inset: BET surface area of samples synthesized at different temperatures) and (**b**) pore size distributions obtained by the BJH (Barrett, Joyner, Halenda) method (inset: Average pore radius of samples synthesized at different temperatures).

The S$_{BET}$ is in the range of 216 to 260 m$^2$ g$^{-1}$, and the highest value corresponds to the sample synthesized at 140 °C (see the inset of Figure 4a). High values of S$_{BET}$ can mean a larger adsorption within the active sites, favoring the photocatalytic efficiency [44,51]. Figure 4b shows the pore size distribution plots, which indicate that the microporous structure coexists with the mesopores for the samples synthesized at temperatures below 160 °C. Meanwhile, the samples T180 and T200 have a uniform pore size, related to a well-defined porous structure. The BET and BJH analysis revealed that an increase in the temperature of synthesis affects not only the surface area but also the pore diameter.

## 2.3. Colloidal Dispersion Properties on the Photocatalytic Activity

The physicochemical properties such as agglomeration, polydispersity, and surface charges of the TiO$_2$ NPs dispersed in an aqueous medium are particularly important for photocatalytic applications. The surface of TiO$_2$ NPs dispersed in water is generally covered by hydroxyl group [52], as shown in Equation (3). The surface charge of TiO$_2$ depends on the solution pH, which is affected by the reactions that occur on the particle surface, as shown in Equations (2) and (3) [53],

$$Ti^{IV} + H_2O \rightarrow Ti^{IV} - OH + H^+ \tag{1}$$

$$Ti^{IV} - OH + H^+ \rightarrow Ti^{IV} - OH_2^+ \tag{2}$$

$$Ti^{IV} - OH \rightarrow Ti^{IV} - O^- + H^+ \tag{3}$$

The stability of colloidal dispersions can be evaluated by measuring the ζ-potential as a function of pH, and from the plots, it is possible to determine the isoelectric point as well as the stable and unstable regions. If all the particles in suspension have a large positive or negative ζ-potential, they will tend to repel each other and there will be no tendency for the particles to agglomerate (stable zones). In contrast, for the zone near the isoelectronic point at zero or low ζ-potential values (unstable zone), there is no repulsion force to prevent the particle agglomeration, and as a consequence, the particles precipitate due to Van der Waals forces. For the TiO$_2$ system, ζ-potential values of approximately ±30 mV demonstrate relatively good dispersion stability (stable zone) of the positively or negatively charged colloidal particles in the pH range, presenting positive values at a low pH and negative values at a high pH.

The surface charge can have a direct effect on the ζ-potential and hydrodynamic diameter of nanoparticles in an aqueous dispersion. An increase/decrease in the ζ-potentials towards positive/negative values enhances the electrostatic repulsion forces, decreasing the D$_H$ of agglomerates and maintaining a monodisperse and more stable dispersion. In

contrast, when the net surface charge lies near zero, van der Waals forces lead to the formation of larger aggregates. In this study, we explore the change in these properties when the as-synthesized TiO$_2$ NPs were dispersed and measure the initial pH (pH$_i$) without acid or base addition. Then, the pH was adjusted with HCl and NaOH for the titration process.

The effect of the dispersion pH on the $\zeta$-potential is shown in Figure 5; this illustrates the titration curves of TiO$_2$ NPs in DI water and in the presence of MB (Figure 5a,b). The point at which the nanoparticle exhibits no net charge is termed the isoelectric point (IEP). As indicated by the dash line in the figure, the IEP is located at pH values ranging from 4.5 to 6 for the TiO$_2$ dispersions. The figure also shows, with an open circle, the point before starting the titration (as-dispersed TiO$_2$ NPs), and its initial pH (pH$_i$) is also indicated. With this, it is also shown that the dispersion pH$_i$ affects the D$_H$ by changing the particle surface charge. Close to the IEP (T160, T180, and T200), significant agglomeration occurs and large flocs are observed, this observation is due to the particle surface charge approaching zero and the attractive Van der Waals forces dominating [54]. However, when pH$_i$ has values in the stable area, the absolute value of the $\zeta$-potential becomes higher, and the D$_H$ becomes smaller (T100, T120, and T140). For our samples, T100 had the smallest hydrodynamic size.

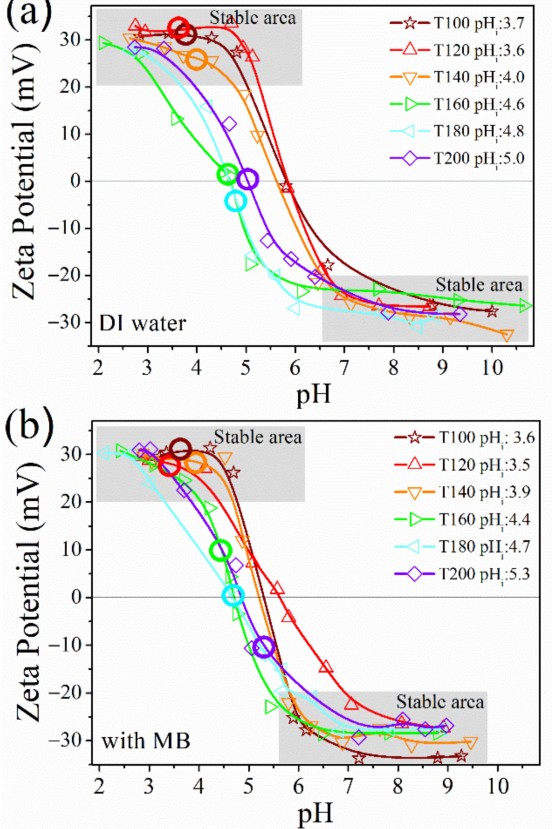

**Figure 5.** The influence of the solution pH on the $\zeta$-potential of TiO$_2$ NP dispersions (**a**) deionized water and (**b**) with methylene blue. The open circle indicates the point before starting the titration (as-dispersed TiO$_2$ NPs), and their initial pH (pH$_i$) is indicated.

The effect of the primary particle size (crystallite size) on the dispersion properties was investigated by dispersing the synthesized TiO$_2$ NPs in DI water and a MB aqueous solution. According to Figure 6a, it was found that the IEP of TiO$_2$ NPs is a function of the primary particle size in both dispersions (DI water and MB). When the primary particle size increases, the IEP decreases, as reported by Suttiponparnit et al. [53]. Knowing that different IEPs can be obtained for the same material depending on the synthesis method and experimental procedure [55–57], we determined that the primary size effect on the dispersion IEP is related to the size of the TiO$_2$ NPs.

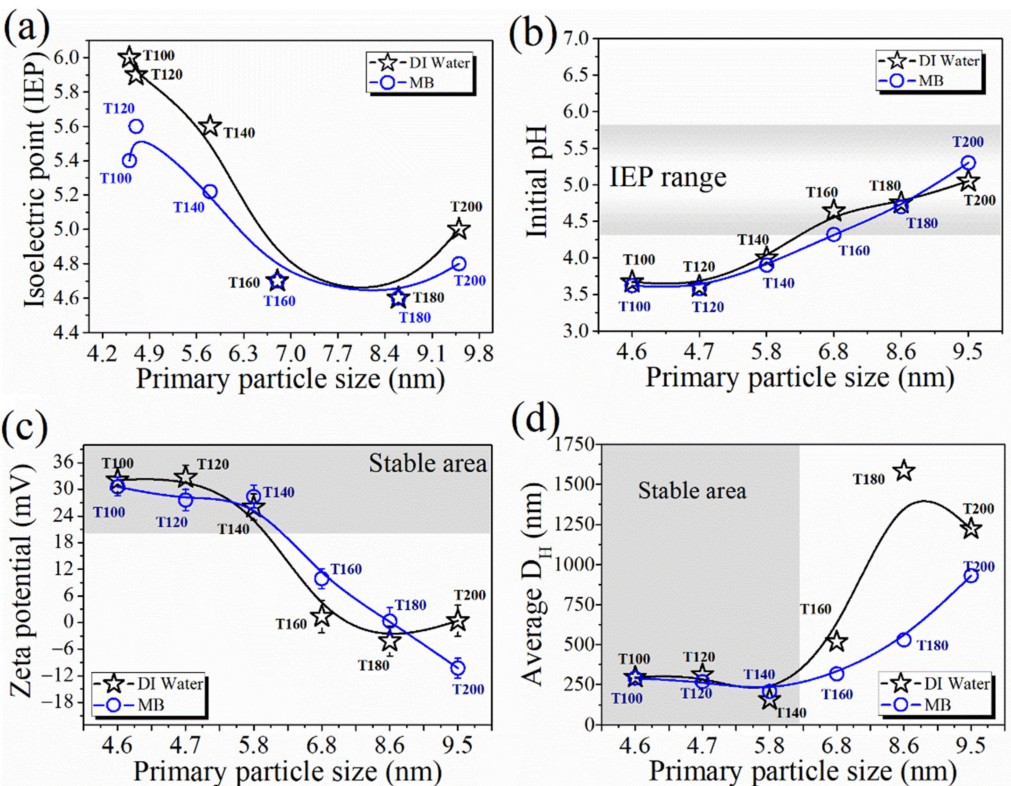

**Figure 6.** The influence of the primary particle size of as-dispersed $TiO_2$ NPs in DI water and with MB on the (**a**) IEP, (**b**) initial pH, (**c**) ζ-potential, and (**d**) average $D_H$ (secondary particle size).

The photocatalytic activity of $TiO_2$ NPs was reported to be size dependent [1,9–12,23,58]. As the nanoparticle size decreases, the percentage of surface atoms/molecules increases significantly, as well as the particle electronic structure, surface defect density, and surface sorption sites [53]. Consequently, the surface reactivity can become dependent on the particle size. Just as the properties of nanoparticles affect the colloidal properties, colloidal properties can affect the performance of other properties such as photocatalysis. In this sense, when the nanoparticles disperse to carry out the heterogeneous photocatalytic reaction, it becomes fundamental to analyze the properties (pH, ζ-potential, and $D_H$ (secondary particle size)) of the $TiO_2$ NP dispersion to know how the photocatalytic process may be affected.

After the $TiO_2$ NPs were dispersed separately in both dispersions (DI water and MB), the $pH_i$ was measured and an increase from 3.6 to 5.3 was observed and approached to the IEP (see Figure 6b). This observation is related to a reduction in the dispersion acidities with the increase in the particle primary size, a behavior that is linked with the surface concentration of –OH groups. When the nanoparticles are dispersed in water, the ionization of the surface -OH groups promotes the liberation of hydronium ions. Consequently, the solution pH decreases as more hydronium ions are generated due to the increase in the titania particle surface area [53].

Figure 6c shows that the dispersion ζ-potential decreased from 35 to 5 mV in the case of DI and from 35 to −10 mV for the MB dispersion with the increase in the primary particle size. This is a consequence of the pH shift, in which more acidic conditions cause an increase in the positive surface charge of the aggregates. Figure 6d indicates that the agglomerate size varies significantly depending upon a variety of factors, including both the pH and nanoparticle chemical composition. As expected, the average $D_H$ is affected due to the IEP; $TiO_2$ NP agglomerates were largest at the IEP. The $D_H$ increased from 200–250 nm to 800–1600 nm because of the associated decrease in the ζ-potential (decrease in repulsive force) favoring agglomeration. The smallest $D_H$ observed was ~200 nm corresponding to T140, presenting a $pH_i$ close to 4.0 and a primary particle size of 5.8 nm. In the case of the

presence of MB, it has only a slight effect on the $pH_i$ and $\zeta$-potential values (Figure 6b,c). However, its effect is more evident in the magnitude of $D_H$ (Figure 6d), which is reduced as a consequence of the increase in the repulsion forces due to the small positive charge induced by the dissociation and adsorption of MB. Nevertheless, the dispersed T140 sample remains as the system with the higher colloidal stability even in the presence of MB. Depending on the interactions in solution, suspended nanoparticles tend to form agglomerates in which the light scattering dramatically depends on the $D_H$, especially in the regime where the agglomerate size is of the order of the wavelength. The results indicate that alterations in the pH have a large effect on the $\zeta$-potential and agglomerate size, which may be used as a predictive measure of the photocatalytic activity of nanoparticles.

### 2.4. Photocatalytic Activity Measurements

In Figure 7, the evolution of the UV-Vis spectrum for MB as a function of time is shown. The dye adsorption and photocatalytic activity of $TiO_2$ NP dispersions were investigated by measuring the change in the relative concentration calculated from the absorbance band at 664 nm of MB as a function of the irradiation time. The slight change in the intensity of the bands observed in the dark (denoted as $-30$ to 0 min in Figure 7) for all samples is related to the adsorption of the dye (4–6%) onto the $TiO_2$ NPs. MB is a cationic dye with a $\lambda_{max}$ value in the visible region at 664 nm and a shoulder at 614 nm, which corresponds to the transition between nonbonding ($n$) to antibonding $\pi^*$ orbitals of the lone pair located on the nitrogen atom of the phenothiazine moiety [59]. Moreover, the characteristic bands for Leuco-methylene blue ($\lambda_{max}$ = 256 nm and 322 nm) are absent [60,61]. Thus, in the experiments, there is no evidence of the formation of the reduced species of $MB^+$. The initial values of the pH (3.5–5.9) prevent the demethylation process of the amino moiety in $MB^+$, which is favored at a basic pH [59].

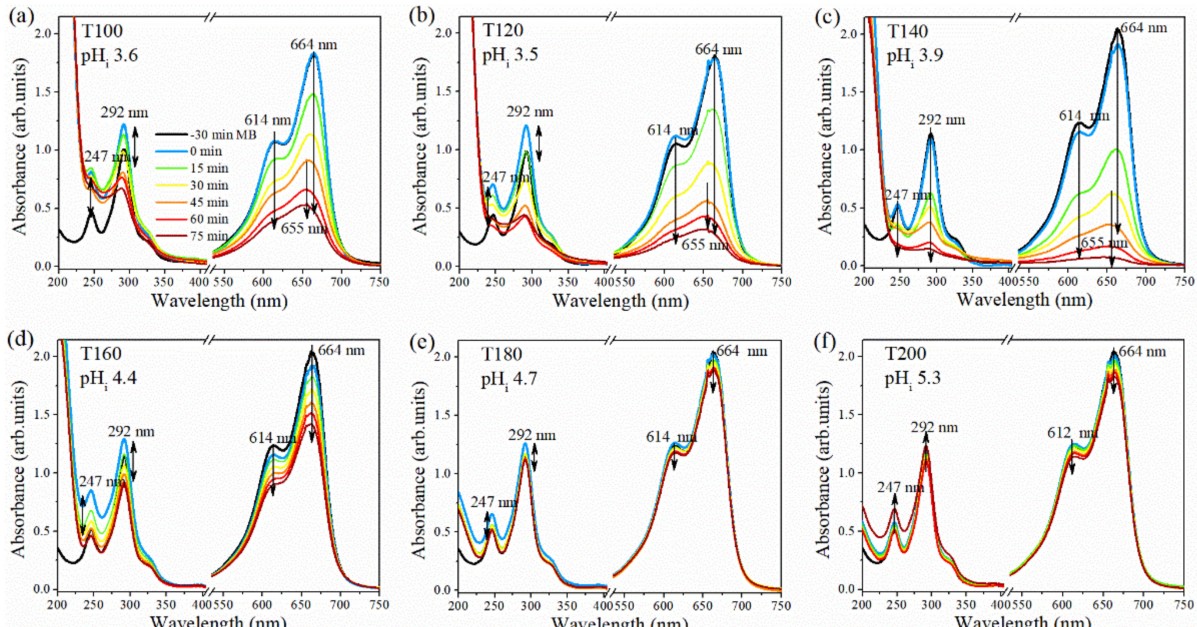

**Figure 7.** Time dependent UV-Vis absorption spectra depicting the degradation of MB solutions catalyzed by micro-mesoporous $TiO_2$ samples: (**a**) T100, (**b**) T120, (**c**) T140, (**d**) T160, (**e**) T180, and (**f**) T200.

Under UV-light irradiation (**denoted as 0 to 75 min in Figure 7**), the $TiO_2$ NPs showed a different photocatalytic activity. The T100, T120, and T140 samples (Figure 7a–c) provoked a dramatic decay in the intensity of the bands at 614 nm and 664 nm, and the T140 sample exhibited the best performance. In the case of the T160 sample (Figure 7d), the intensity decrease of the bands is less pronounced. In contrast, for the T180 and T200 samples

(Figure 7e,f), the change is practically negligible. In order to give more insight into the photodegradation process using the T140 sample, the spectra of MB in the dark and after 75 min of UV irradiation are presented in Figure 8. The total disappearance of the absorbance bands located at 247 nm and 292 nm because of the photocatalytic reaction can be seen. These bands correspond to transitions between bonding ($\pi$) to antibonding ($\pi^*$) orbitals of the aromatic rings of the phenothiazine moiety. When the phenothiazine moiety is broken by the radical hydroxyl attack, this leads to the formation of phenolic derivates, which in turn can be degraded to simple organic compounds [62,63].

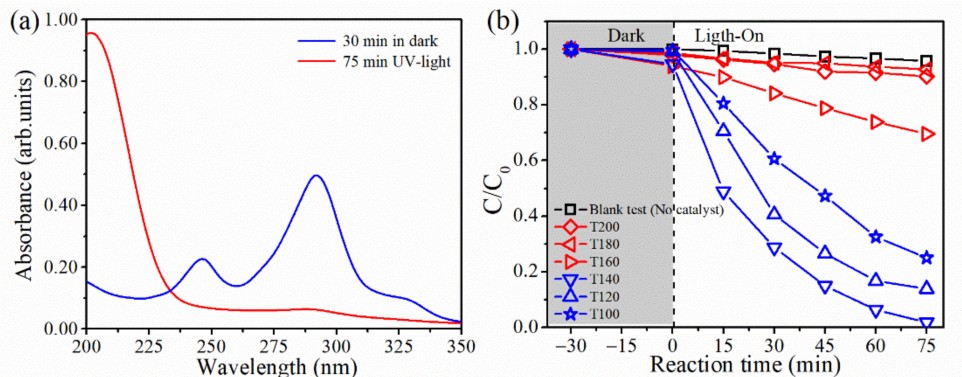

**Figure 8.** (**a**) UV spectrum (200–350 nm) for a dispersion containing the T140 sample in dark (blue line) and after 75 min of UV irradiation (red line). (**b**) Time dependence of the relative concentration changes of MB and UV irradiation time of all samples.

A comparison of the time dependence of the relative concentration ($C/C_0$) for all samples is shown in Figure 8b, and the photodecomposition rate constants obtained from this analysis are shown in Table 2. In the plots of $\ln(C/C_0)$ versus UV irradiation time (Figure A3), straight lines were found for all samples, indicating that the mechanism of degradation of MB for T100, T120, and T140 NPs follows the classically expected pseudo-first-order kinetics [52], while for T160, the degradation mechanism lies in an intermediate behavior that seems to be a competition between photocatalyzed degradation and direct photolysis. For comparison, the photocatalytic degradation of MB by direct photolysis was only ~5% over 75 min. Meanwhile, by using the T180 and T200 samples, the MB decomposition appears to take place via direct photolysis indicative of low photocatalytic activity.

**Table 2.** Rate constants for the degradation of MB using TiO$_2$ NPs obtained at different temperatures.

| Sample | Rate Constant ($10^{-2}$ min$^{-1}$) | TOR$_s$ ($10^{-4}$ M$^{-1}$ min$^{-1}$ m$^{-2}$ g$^1$) | Conversion (%, 75 min) | $R^2$ |
|---|---|---|---|---|
| T100 | 1.88 | 0.82 | 75 | 0.99 |
| T120 | 2.55 | 1.20 | 86 | 0.99 |
| T140 | 5.03 | 1.93 | 98 | 0.98 |
| T160 | 0.41 | 0.17 | 26 | 1.00 |
| T180 | 0.08 | 0.03 | 6 | 0.96 |
| T200 | 0.11 | 0.05 | 8 | 0.97 |

The best photocatalytic activity corresponds to the T140 sample with a degradation of 98% after 75 min ($5.03 \times 10^{-2}$ min$^{-1}$), followed by T120 (86%) and T100 (75%). In contrast, for the T160, T180, and T200 samples, the relative concentration presents a quasilinear time dependence (black test in Figure 8b) [64] and presented a low degradation of only 6–26% after 75 min. It is important to question the factors that define the photocatalytic performance for the materials under study.

The SEM, HRTEM (Appendix A Figure A2), and DLS results help us to explain the relationship between the microstructure of the samples and the aggregation when

crystallites are dispersed in water. Furthermore, we have added the comparison between the degradation percent with crystallite size, measured band gap, surface area, and atomic percent of surface OH groups from XPS.

If we try to get first insight into the correlation between the photocatalytic performance and the material structural properties, we should retake the primary attributes of the photocatalyst such as crystallinity, crystallite size, optical response, and specific surface areas [1,9–12,23,58]. Figure 9a–d show a comparison between the degradation percent with crystallite size, measured band gaps, surface area, and atomic percent of surface OH groups. We found that $TiO_2$ NPs with smaller crystallites (4–7 nm) and a higher percent of surface OH groups (15–22%) show a better photocatalytic performance for MB degradation (Figure 9a,b). In particular, the XPS results show that the hydroxyl concentration (atomic percent of -OH at 530.6 eV) decreases gradually with the increase in the temperature of synthesis (inset Figure 9b). On the other hand, the degradation percent is independent of the measured band gap, and all bandgap measurements in this work refer to the optical bandgap. Bandgap energy values of 3.21–3.26 $\pm$ 0.05 eV are obtained for direct transitions, which agree well with the values reported for anatase [65,66]. Therefore, we have not found evidence of the quantum confinement effect on the energy gap. Note that the bandgaps remain practically constant with the temperature of synthesis (Figure A4). Therefore, no evidence that the difference in the photocatalytic activity may be due to optical properties is found (Figure 9c). In the case of the surface area, there is not a clear correlation with the measured degradation percent. Usually, a larger surface area is associated with a larger number of surface sites for degradation. However, Figure 9d shows that even samples with similar surface areas can show a dramatic difference in the photodegradation. Thus, to complement the understanding of the observed trends in the photodegradation process, it is necessary to include the description of the surface chemistry in the colloidal dispersion.

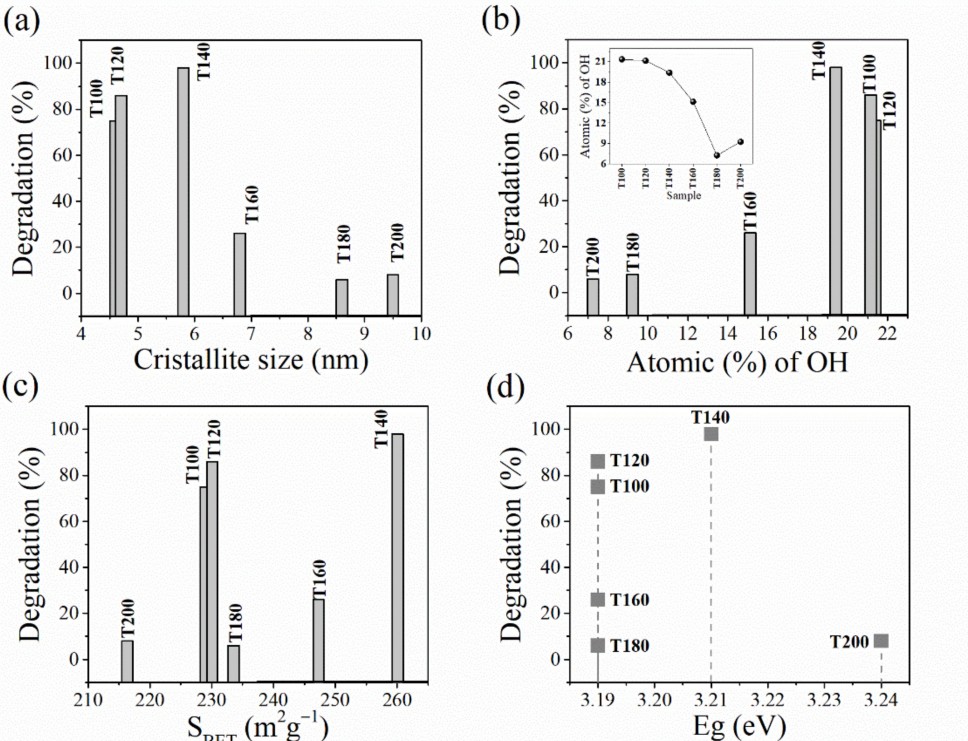

**Figure 9.** Analysis of degradation (%) in terms of the structural properties: (**a**) crystallite size, (**b**) atomic (%) of OH groups; inset variation of atomic (%) OH for each sample, (**c**) surface area-BET, and (**d**) Eg (eV).

With this idea in mind, in Figure 10a–c, we plotted the degradation percentage in terms of the $pH_i$ of the $TiO_2$ dispersions. The $TiO_2$ dispersions with $pH_i$ values between

3.4 and 4.0 displayed the highest photocatalytic performance, in contrast to those with $pH_i$ between 4.4 and 5.3 (Figure 10a). In Figure 10b, the T100, T120, and T140 samples showed photodegradation rates of 1.88, 2.55, and $5.03 \times 10^{-2}$ min$^{-1}$, respectively, while for the T160, T180, and T200 samples, the photodegradation rate dropped by one order of magnitude. In the inset of Figure 10b, the zeta potential of the TiO$_2$ dispersions as a function of $pH_i$ is plotted. From this plot, it is evident that the samples with the higher photocatalytic activity are positively charged ($\zeta$ = 25 to 30 mV), whereas the lowest degradation is observed in samples where the $\zeta$-potential approaches zero. Finally, in Figure 10c, the average $D_H$ is plotted versus the initial pH. As we can see, $D_H$ has its minimum value (190 nm) at $pH_i$ = 3.9, corresponding to the T140 sample; meanwhile, for $pH_i$ larger than 4, the T160, T180, and T200 samples form large aggregates ($D_H$ of 300–850 nm). The formation of large aggregates can be explained in terms of the interplay of the electrostatic repulsion and the van der Waals forces between TiO$_2$ NPs. As the pH approaches IEP, the $\zeta$-potential of the nanoparticles is near zero, minimizing the electrostatic repulsion and leading to an unstable dispersion and diminishing the photocatalytic performance. For T100–T140 NPs, a large positive surface charge guarantees electrostatic repulsion, favoring dispersion stability.

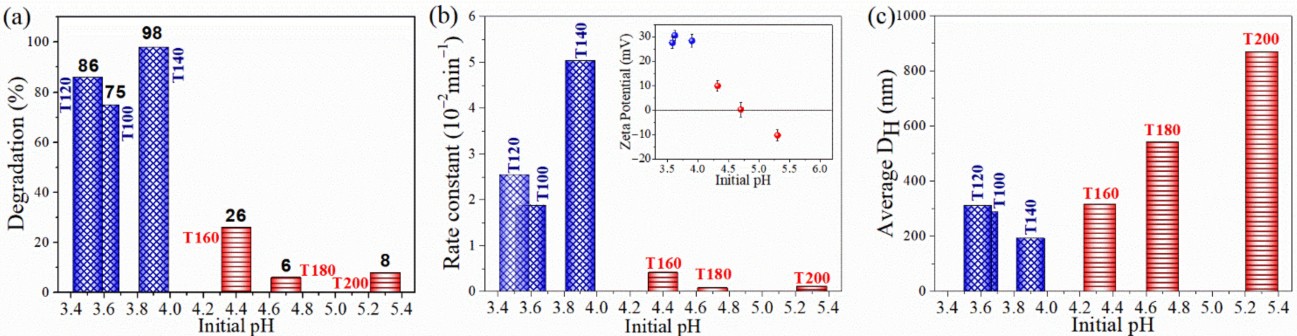

**Figure 10.** (**a**) Distribution of the degradation percentage in terms of the initial pH, (**b**) rate constant against the initial pH (the inset: $\zeta$-potential as a function of the initial pH), and (**c**) the average $D_H$ plotted against the initial pH of the TiO$_2$ dispersions.

In heterogeneous photocatalysis, it is important to normalize the rate constants with respect to the surface area [67] of the samples. As it can be observed in column three (Table 2), the same trend of the photocatalytic performance remains after normalization. The rate constants can be interpreted now as a turnover rate (TORs). Figure 10 shows the tight relationship between the surface charge of TiO$_2$ NPs and the photodegradation of MB in an aqueous solution. In the case of the systems under study, we have found that in acidic media, the photodegradation rate is optimal for a pH value of around 4 where the NPs are positively charged ($\zeta$ = 30 mV) with $D_H$ around 190 nm. In contrast, for dispersions containing T160, T180, and T200 NPs, the TORs decreases when the hydrodynamic diameter increases (300–850 nm). This behavior suggests that light scattering could be playing a predominant role in the photodegradation of MB [36]. In order to gain further insight into this behavior, we have explored the theoretical optical response of our TiO$_2$ NP dispersions in the framework of the Mie's theory. Because we do not have access to the actual fractal structure of the agglomerates within the dispersion, it was necessary to adopt the assumption that we are dealing with spherical homogeneous particles whose radius is known from DLS measurements. This assumption has previously been successful in explaining the photocatalytic activity behavior of hollow TiO$_2$ microspheres [68].

The calculated absorption, scattering, and extinction cross sections as a function of the secondary particle sizes are shown in Figure 11a. The cross-sections were calculated at a wavelength of 254 nm (the same as the UV-lamp), and the dielectric constant values were 2.48 for titania and 1.33 for water. Note that these results are in good agreement with previous estimations [36]. It is interesting to note how the Mie theory predicts that for TiO$_2$ NPs, the scattering surpasses the absorption component when the particle size

is larger than 55 nm (see inset in Figure 11a). According to DLS, our sample dispersions consist of secondary particle sizes that are larger than this value; therefore, it is expected that light scattering adversely impacts the quantum efficiency by removing energy from the incident light source (beam attenuation). The total absorption cross section of each sample dispersion can be estimated by multiplying the single particle cross section by the particle number (calculated with the average secondary particle size obtained from the DLS measurements).

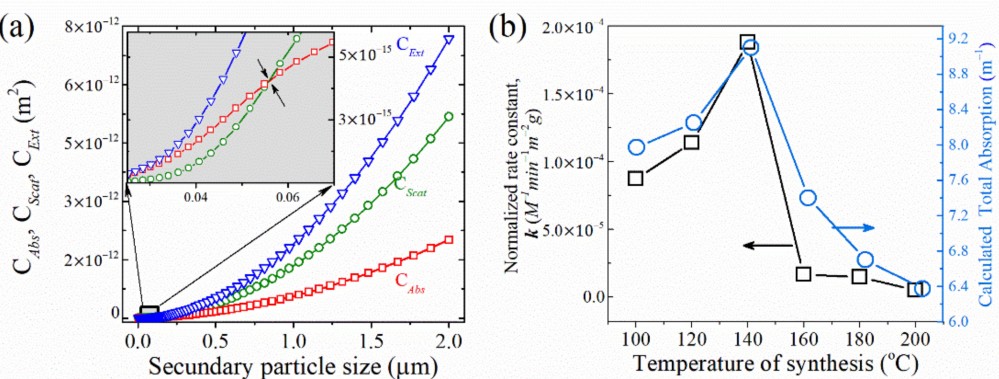

**Figure 11.** (**a**) Scattering, absorption, and extinction cross sections as a function of the particle size, calculated according to the Mie's theory. The inset corresponds to a zoom at smaller particle sizes to show the crossover of the absorption and scattering coefficients. (**b**) Normalized rate constants (TORs) of photocatalysis and calculated total absorption coefficient for the $TiO_2$ NPs synthesized at different temperatures (100–200 °C).

The normalized rate constants (TORs) and the calculated total absorption cross sections for the NPs synthesized at different temperatures are compared in Figure 11b. The results in Figure 11b suggest a strong correlation between the photocatalytic performance and the total absorption of $TiO_2$ NPs. We can observe that the dispersion containing T140 with a maximum absorption exhibits the maximum normalized rate constant. In contrast, lower values of absorption cause a reduction in the rate constant (TORs). With the aim to examine the correlation between the calculated total absorption (TA) and the TORs, we resort to a linear regression analysis with a null hypothesis ($H_0$) in such a way that the linear relationship TA vs. TORs has a slope of 0, implying that variations in the TA of NPs cannot explain the variation of TORs for the systems under study. The calculated *p*-value from F-statistic provides a way for globally testing if the independent variable (TA) is related to the response variable (TORs). If such an analysis does not validate $H_0$, it can readily allow the rejection of $H_0$ when the *p*-value is a small number, i.e., when the *p*-value < $\alpha$, being $\alpha$ = 0.01, 0.05, or 0.1 depending on the selected significance level. In this case, we have selected a significance level $\alpha$ = 0.05 (95% confidence interval) for testing $H_0$. The small *p*-value obtained (0.003) for the global test of regression (Table 3) is lower than the significance level, clearly showing that $H_0$ can be rejected, and as a consequence, the total absorption is able to explain the variation in TORs for the systems under study (within a 95% confidence interval). Furthermore, the value of 0.90 for R-multiple (Table 3) shows a strong linear dependence between TA and TORs. The computed square-R value indicates that the total absorption can explain 95% (Table 3) of the variation for TORs. In the case of the linear regression coefficients, the computed *p*-values for the intercept and slope in Table 3 confirm that changes in the total absorption of NPs are related to changes in the response variable (TORs).

**Table 3.** Parameters for the global (F-test) and individual coefficient (*t*-test) for linear regression at a significance level $\alpha = 0.05$.

| Global Test | | | | |
|---|---|---|---|---|
| **R-multiple** | **Square-R** | **Std. Error** | **F** | ***p*-Value** |
| 0.90 | 0.95 | 0.00002 | 39.89 | 0.003 |
| **Coefficient test** | | | | |
| | Std. error | Coefficients | *p*-value | *t*-test |
| **Intercept** | 0.000082 | −0.00044 | 0.005 | −5.40 |
| **Slope** | 0.000010 | 0.00006 | 0.003 | 6.31 |

This interesting result can be explained by the Grotthus–Draper law from which it is expected that only radiations that are absorbed by the reacting system are effective in producing a chemical change [69]. Thus, the total absorption (which is related to the secondary particle) can be adjusted by modifying the pH. It was observed that for the $TiO_2$ NPs under study, an eventual increase in the pH (4–5.5) reduces the effective surface charge and dispersion stability, which has a decrease of one order of magnitude on rate constants ($10^{-3}$ min$^{-1}$) for $TiO_2$ agglomerates with a larger $D_H$ (300–850 nm) and with a lower total absorption coefficient.

## 3. Materials and Methods

### 3.1. TiO$_2$ NP Powder Synthesis and Characterization

A volume of 21 mL of titanium tetraisopropoxide (97%, Aldrich, Oakville, Canada) was added dropwise to 4 mL of glacial acetic acid under constant stirring. The resulting clear sol was added dropwise to 145 mL of an aqueous solution $15 \times 10^{-4}$ M of cetyltrimethylammonium bromide (CTAB) (BioXtra, ≥99%, Aldrich) under stirring. Subsequently, the solution was stirred continuously for 1 h and aged for 75 min at 80 °C (peptized). Then, 25 mL of the previous solution was transferred into a 30 mL quartz reaction vessel and then irradiated with a microwave system (Monowave 300 Anton Paar, Graz, Austria) at different temperatures (100, 120, 140, 160, 180, and 200 °C) for 5 min under stirring at 400 rpm [42]. After centrifugation at 6000 rpm for 30 min, the precipitate was washed twice with deionized water (17 MΩ•cm) and ethanol. Then, the paste was dried (vacuum at 80 °C for 12 h). Samples were identified as TXXX where XXX refers to the temperature of synthesis (e.g., T140).

The crystal phase of the $TiO_2$ materials was analyzed by X-ray diffraction (XRD) with a Siemens D-5000 with Cu K$_\alpha$ radiation at 34 KV and 25 mA in the range of 5–90° (2θ) with a 0.02° step size and 1 s integration time. The crystallite size was determined using the program Bruker's Topas-4. The samples were also characterized by Raman spectroscopy with a Raman confocal WiTec alpha 300 (at 488 nm, 100× and grating of 1800 grooves/mm). Morphological features of the as-prepared materials were studied with a scanning electron microscope (FE-SEM JEOL JSM-6700F, Akishima, Japan) and high-resolution transmission electron microscopy (HR-TEM) using a JEM-ARM200F-JEOL instrument in HAADF-STEM and BF-STEM modes. Primary particle size distributions were acquired from several observation fields of HR-TEM. An Ocean Optics (USB 2000) fiber optics spectrophotometer was used for UV-Vis diffuse reflectance characterization. The reflectance spectra were analyzed using Kubelka–Munk formalism to convert the reflectance, R, into the equivalent absorption coefficient, $F(R) = (1 - R)^2/2R$. The band gaps ($E_g$) were derived from extrapolating the linear region near the absorption threshold. The binding energy (BE) was identified by X-ray photoelectron spectroscopy (XPS, Surface Analysis, Thermo Scientific, Waltham, United States) with a monochromatic Al-K$\alpha$ X-ray source (1486 eV) under ultrahigh vacuum conditions ($<10^{-8}$ Torr). The emitted photoelectrons were sampled from an area of 600 μm$^2$. The C 1s, O 1s, and Ti 2p peak components are noticeable in the wide scan spectrum of all samples. The adventitious C 1s peak at 284.8 eV was used as an internal standard of reference, and the measurements was made without an Argon ion

beam to prevent the reduction of $Ti^{4+}$. Nitrogen adsorption–desorption isotherms of the materials were determined at $-196$ °C using an instrument (Belsorp, BEL JAPAN Inc., Osaka, Japan). Each sample was degassed at 100 °C for 24 h under $10^{-5}$ Pa. Surface areas were calculated by the Brunauer–Emmett–Teller (BET) method in the relative pressure range of $P/P_0 = 0.05$–0.25. The average pore sizes and pore size distributions were obtained through the Barrett–Joyner–Halenda (BJH) method.

### 3.2. Characterization of TiO₂ Dispersions

The $D_H$ and $\zeta$-potential of $TiO_2$ NPs dispersions were characterized using the ZetaSizer Nano ZS 3600 (Malvern Instruments Inc., Malvern, United Kingdom) utilizing DLS and electrophoretic light scattering (ELS), respectively [70]. DLS measures the intensity of the laser light that is scattered from dissolved macromolecules or suspended particles. The dispersion $D_H$ is derived from the temporal evolution of the scattered light intensity using the Stokes–Einstein equation [71]. ELS determines the frequency or phase shift of an incident laser beam produced by electric field driven particle migration, reported as the electrophoretic mobility. Particle $\zeta$-potential is calculated from the measured electrophoretic mobility using the Smoluchowski equation [71,72]. The agglomeration of $TiO_2$ NPs can be generated by the pH change. To examine the effect of pH on the $D_H$, $\zeta$-potential, and isoelectric point (IEP) of colloid $TiO_2$ NPs (in DI-water and DI-water/MB), the pH was adjusted by adding HCl and NaOH. Typically, colloidal dispersion with $TiO_2$ NPs and of MB at the same concentrations of photocatalytic test was used. Therefore, pH, $D_H,$ and $\zeta$-potential were also measured in DI-water and DI-water/MB. In all measurements, colloid titania NPs were stirred for 30 min before the DLS and ELS measurement. All measurements were carried out at $25 \pm 1$ °C and $\lambda = 633$ nm, which was maintained by the Zetasizer instrument, Malvern, United Kingdom (Malvern Instruments Zen 3600 Zetasizer). The repeatability of the $D_H$ and the $\zeta$-potential was verified with ten measurements. It should be noted that the pH of the dispersions in the photocatalytic activity test was not modified. IEP of the $TiO_2$ NPs was obtained from the $\zeta$-potential versus pH curves.

For optical cross-section calculations, first Mie scattering and absorption cross sections were calculated using the Mieplot software, which is a user-friendly interface that implements the algorithm of Bohren and Huffmann for Mie scattering equations [73] of a homogeneous sphere. The relevant empirical parameters for cross-section calculations are the dielectric constant of the media (water) and dielectric sphere ($TiO_2$) and the nanoparticle size in micrometers. For the dielectric constant of water, a value of 1.33 was used, and for $TiO_2$, a value of 2.48 was used. These values were taken from the literature [74]. The nanoparticle size in micrometers was obtained directly from the hydrodynamic diameter of the DLS measurements. For each particle size, the optical cross-sections were calculated for wavelengths between 200 and 400 nm; however, with the aim to evaluate the influence of optical absorption and scattering under photocatalytic conditions, we only compare the cross-sections at 254 nm, which is the wavelength of the UV-lamp used in the photocatalytic experiments.

### 3.3. Photodegradation of MB

The photocatalytic activity of $TiO_2$ NPs was evaluated by measuring the degradation reaction of MB under UV illumination at regular intervals. MB is based on a tricyclic phenothiazine chromophore and is an intensely colored blue cationic dye ($MB^+$). In an aqueous solution, $MB^+$ has two characteristic UV-Vis absorption bands at 292 nm and 664 nm, which correspond to the $\pi$-$\pi^*$ and $n$-$\pi^*$ transitions, respectively. Often, the degradation rate of $MB^+$ is determined from the decrease in the absorbance band at 664 nm.

In each experiment, prior to UV irradiation, the reaction dispersion was magnetically stirred in the dark for 30 min until adsorption/desorption equilibrium was reached. The adsorption of MB on $TiO_2$ samples was measured before UV irradiation. The adsorption and photo-discoloration of the samples were investigated in a lab scale photocatalytic reactor. The reactor consisted of a 250 mL beaker and a UV-light lamp (Pen-Ray PS1 model,

UVP-11SC-1, typical intensity 254 nm @ 1.9 cm is 4.1 mW cm$^{-2}$) as the light source to trigger the photo-discoloration.

The concentration of MB was calculated by the Lambert–Beer law using a spectrophotometer at 664 nm. The discoloration rate of MB was calculated by the following equations [17,75]:

$$W \text{ (adsorption)} = \frac{C_0 - C_1}{C_0} \times 100 \tag{4}$$

$$W \text{ (degradation)} = \frac{C_1 - C_2}{C_1} \times 100 \tag{5}$$

where $C_0$ is the initial dye concentration, $C_1$ is the concentration after adsorption equilibrium, and $C_2$ is the concentration after a certain time of irradiation. The colloidal TiO$_2$ NPs with MB dispersion (V = 200 mL, $C_0$ of MB = $1 \times 10^{-2}$ g/L, 0.1 g of TiO$_2$) were mixed in the reactor. To follow the MB concentration as a function of time, 2 mL of the suspension was taken out and filtered through a Millipore filter (pore size of 0.45 μm) to remove large agglomerates and the filtered solution was used to measure the absorbance. An aliquot of 2 mL was taken every 15 min for a total of 75 min of irradiation.

## 4. Conclusions

In the present study, the relationship between the photodegradation rate of the MB dye and the effective surface charge of TiO$_2$ NP dispersions was addressed. TiO$_2$ NP dispersions were prepared from porous anatase/brookite nanoparticles (~10% brookite) synthesized by microwave heating at different temperatures (100–200 °C) in one step and a brief period of time (5 min). The analysis of XRD and HRTEM shows that our samples have average primary sizes in the range of 4–10 nm that were controlled by changing the reaction temperature. The properties of MB dispersion containing TiO$_2$ NPs (D$_H$, pH, $\zeta$-potential, and IEP) were found to be strongly dependent on the microwave synthesis conditions. Characterization of the TiO$_2$ dispersion properties in MB-containing samples shows that acidic dispersions (pH = 3.6–4.0) with D$_H$ ranging from 200–250 nm and positive zeta potential ($\zeta$ = 35–30 mV) exhibit large stability and a pseudo-first-order kinetics with degradation rate constants of the order of $10^{-2}$ min$^{-1}$. The higher rate constant ($5.03 \times 10^{-2}$ min$^{-1}$) corresponded to 98% of degradation efficiency within 75 min for the MB dispersion containing TiO$_2$ NPs synthesized at 140 °C. This enhanced rate is a consequence of a synergic effect between the large surface area (260 m$^2$ g$^{-1}$), surface charge of TiO$_2$ NPs, and the generation of hydroxyl free radicals. It was observed that the eventual increase in the pH (4–5.5) reduces the effective surface charge and dispersion stability, with a decrease of one order of magnitude in the rate constants ($10^{-3}$ min$^{-1}$) for TiO$_2$ agglomerates with a larger hydrodynamic diameter (300–850 nm). The analysis of the total absorption cross-sections as a function of D$_H$ (framework of Mie scattering theory) predicts that dispersions with larger agglomerates have lower values of total absorption, which correlates with lower measured photodegradation rates. According to our results of the dispersion properties, pH$_i$ can change the dispersion state by altering the $\zeta$-potential (surface charge). For instance, bringing the pH close to the nanoparticle IEP will enhance agglomeration and result in larger hydrodynamic sizes. The photocatalytic performance depends on the initial pH and dispersion properties of the TiO$_2$ NPs, demonstrating the necessity of addressing the dispersion properties of TiO$_2$ NP aggregates when dealing with photocatalysis by semiconductor nanoparticles in a colloidal dispersion. These results have important implications for the performance of photocatalytic studies, such as the preparation of nanoparticle dispersions for photocatalytic tests and the interpretation of optical absorption.

**Author Contributions:** Study design and literature search M.C.C.-C., C.M.R.-C. and M.Á.R.-G.; artwork and figures M.C.C.-C.; software, C.M.R.-C. and G.R.-G.; writing—original draft preparation, M.C.C.-C. and G.R.-G.; writing—review and editing, M.Á.R.-G., J.V.-C. and M.R.-P. All authors have read and agreed to the published version of the manuscript.

**Funding:** This research was funded by grants FOMIX- YUCATAN 170120, FOMIX-YUCATAN 2008-108160, and CONACYT LAB-2009-01-123913, 292692, 294643, 188345. CONACYT, SENER, and CICY for funding through the Renewable Energy Laboratory of Southeast Mexico (LENERSE; Project 254667), as well as the CONACYT Infrastructure Project 2013-204822, CONACYT under grants in the Frontier Science project, FORDECYT-PRONACES CF/2019/848260, CONACYT Postdoctoral fellowship: I1200/224/2021.

**Institutional Review Board Statement:** Not applicable.

**Informed Consent Statement:** Not applicable.

**Data Availability Statement:** Not applicable.

**Acknowledgments:** XRD: BET, DLS, and XPS measurements were performed at the National Laboratory for the Study of Nano and Biomaterials (LANNBIO) at CINVESTAV-Mérida. The authors would like to thank Patricia Quintana for access to LANNBIO; Daniel Aguilar, William Cauich, Dora A. Huerta, Beatriz Heredia, Leny Fernanda Pinzón Espinosa, William Santiago González Gómez for their technical help and Mario Herrera Salvador for the corrective maintenance of the D-8 Advance diffractometer. HRTEM measurements were performed at the Advanced Laboratory of Electron Nanoscopy (LANE) at CINVESTAV-México. M.C.C.C acknowledges CONACYT for the financial support in the realization of the postdoctoral research at Autonomous University of Campeche (UAC).

**Conflicts of Interest:** The authors declare no conflict of interest.

## Appendix A

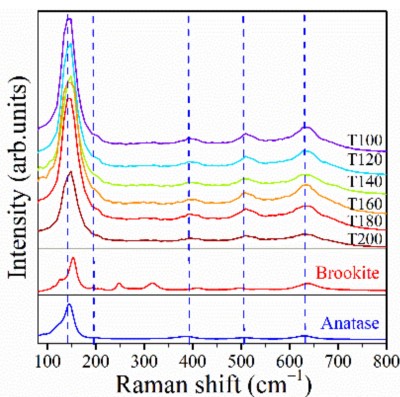

**Figure A1.** Evolution of the Raman spectra of samples obtained at 100–200 °C for all TiO$_2$ NP samples.

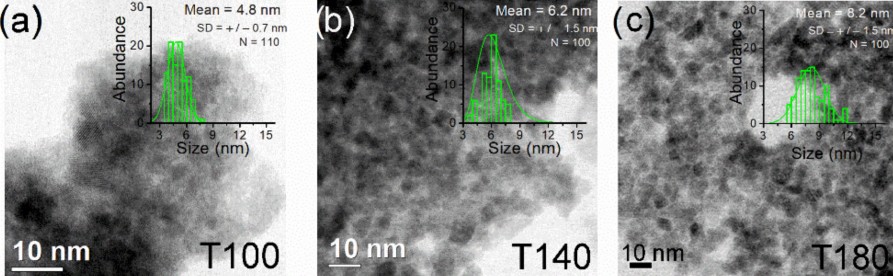

**Figure A2.** Selected HRTEM micrographs of the particle size evolution as a function of the synthesis temperature, (**a**) 100, (**b**) 140, and (**c**) 180 °C. Insets of each micrograph corresponds to the particle size distribution determined from several images, from N = 100.

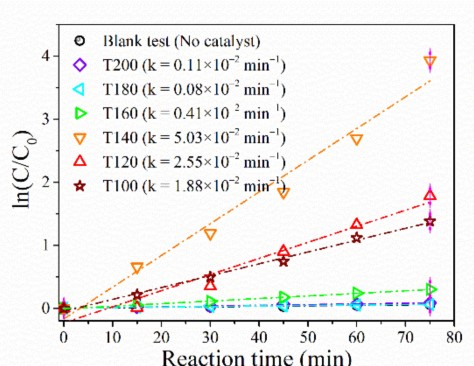

**Figure A3.** Plot of $\ln(C/C_0)$ with UV irradiation time for the photocatalysis of MB by $TiO_2$ samples.

In Figure A4, values of the gap energy for direct transitions derived from the reflectance spectra of all $TiO_2$ NP samples are shown. Figure A4 inset shows the $(F(R)h\nu)^2$ against photon energy (eV) for a direct transition, the intercept of the tangent $(F(R)h\nu)^2 = 0$ of the Tauc plot gives a band-gap energy of 3.21 eV for T140.

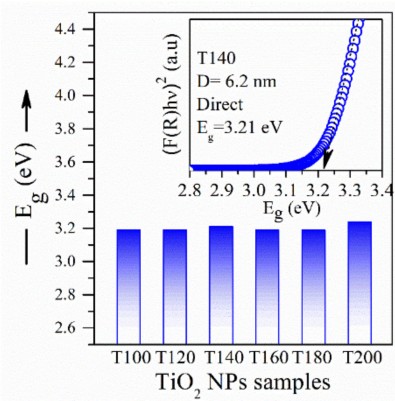

**Figure A4.** Values of the gap energy for direct transitions derived from the reflectance spectra of all $TiO_2$ NP samples. Inset: Analyses of the electronic absorption spectrum for T140 assuming direct transitions.

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
