# Peer review of "Synergistic Correlation in the Colloidal Properties of TiO2 Nanoparticles and Its Impact on the Photocatalytic Activity"

_inorganics, doi:10.3390/inorganics10090125_

Round 1
Reviewer 1 Report
This work is worth doing, and the authors present some new results in the paper. However, the manuscript should be significantly revised before resubmission. I would like to make the following comments.
1) The title should be rewritten to meet scientific publication requirements.
2) The English of the full text needs to be improved. For example, in the abstract, " From TiO2 NPs (4-10 nm) were prepared colloidal dispersions for heterogenous photocatalysis test and its properties (pH, hydrodynamic diameter (DH), zeta potential (ζ-potential) and isoelectric point (IEP)) were study and compared with dispersion without MB." and so on.
3) Why are there no SEM/HRTEM measurements in this work? Usually, detailed results from SEM/HRTEM are required to understand the structural information of nanomaterials.
4) The discussion is too simplistic and general. As a research article, the authors should make discussion in close relation with their measured and/or analyzed results in the paper.
5) The literature is too old. The authors should add some relevant literature published in the past three years.
Reviewer 2 Report
The paper by Ceballos-Chuc et. al. describes the synthesis and characterization of TiO2 dispersions with varying temperatures. The photocatalytic degradation of methylene blue was investigated with TiO2 dispersion as photocatalysts. The author concludes that the zeta potential (highly dependent on pH value) can significantly affect the degradation rate. However, the author didn’t correlate with the structure parameter of TiO2 and photocatalytic property. This reviewer suggested carefully analyzing the structure of synthesized TiO2 dispersion and discussed the relationship between samples structure and physical properties (BET, zeta potential, pore size…).
In conclusion, this reviewer thinks this paper should be major revised before suitable for publication.
Reviewer 3 Report
The authors did a great job demonstrating how the initial pH, hydrodynamic diameter, specific surface area, and isoelectric point play roles in the photocatalytic methylene blue degradation by TiO2.
All the characterizations were performed and interpreted very in-depth. There are very few points that need to be improved before this manuscript can be considered for publication. So, I all list them:
1st – In Figures 4 and 5 some pH ranges are described as stable regions. As far as I understand, the stable regions are the ones far enough from the isoelectric point, so particle aggregation is prevented in these regions. However, other readers may not be able to make this connection between this concept and the expression stable region. So, I recommend the authors to add some sentences explaining exactly what the stable region means.
2nd – About the results presented in Figure 9, the authors need to add a section in experimental procedure section explaining in details how they calculated the absorption, scattering and extinction cross-sections as function of the secondary particle sizes. For instance, which equations, constants, and assumptions were used to accomplish these results.
Round 2
Reviewer 2 Report
The authors have significantly revised and improved the quality of the manuscript , including a lot more in-depth discussion. I suggest to accept the paper under current form.